# Dosimetry of [^212^Pb]VMT01, a MC1R-Targeted Alpha Therapeutic Compound, and Effect of Free ^208^Tl on Tissue Absorbed Doses

**DOI:** 10.3390/molecules27185831

**Published:** 2022-09-08

**Authors:** Kelly D. Orcutt, Kelly E. Henry, Christine Habjan, Keryn Palmer, Jack Heimann, Julie M. Cupido, Vijay Gottumukkala, Derek D. Cissell, Morgan C. Lyon, Amira I. Hussein, Dijie Liu, Mengshi Li, Frances L. Johnson, Michael K. Schultz

**Affiliations:** 1Viewpoint Molecular Targeting, Inc., Coralville, IA 52241, USA; 2Invicro, LLC, Needham, MA 02494, USA; 3Department of Radiology, The University of Iowa, Iowa City, IA 52242, USA; 4Department of Radiation Oncology, The University of Iowa, Iowa City, IA 52242, USA; 5Departments of Radiology and Radiation Oncology, The University of Iowa, Iowa City, IA 52242, USA

**Keywords:** ^212^Pb, ^203^Pb, ^208^Tl, MC1R, dosimetry, absorbed dose, melanoma

## Abstract

[^212^Pb]VMT01 is a melanocortin 1 receptor (MC1R) targeted theranostic ligand in clinical development for alpha particle therapy for melanoma. ^212^Pb has an elementally matched gamma-emitting isotope ^203^Pb; thus, [^203^Pb]VMT01 can be used as an imaging surrogate for [^212^Pb]VMT01. [^212^Pb]VMT01 human serum stability studies have demonstrated retention of the ^212^Bi daughter within the chelator following beta emission of parent ^212^Pb. However, the subsequent alpha emission from the decay of ^212^Bi into ^208^Tl results in the generation of free ^208^Tl. Due to the 10.64-hour half-life of ^212^Pb, accumulation of free ^208^Tl in the injectate will occur. The goal of this work is to estimate the human dosimetry for [^212^Pb]VMT01 and the impact of free ^208^Tl in the injectate on human tissue absorbed doses. Human [^212^Pb]VMT01 tissue absorbed doses were estimated from murine [^203^Pb]VMT01 biodistribution data, and human biodistribution values for ^201^Tl chloride (a cardiac imaging agent) from published data were used to estimate the dosimetry of free ^208^Tl. Results indicate that the dose-limiting tissues for [^212^Pb]VMT01 are the red marrow and the kidneys, with estimated absorbed doses of 1.06 and 8.27 mGy_RBE = 5_/MBq. The estimated percent increase in absorbed doses from free ^208^Tl in the injectate is 0.03% and 0.09% to the red marrow and the kidneys, respectively. Absorbed doses from free ^208^Tl result in a percent increase of no more than 1.2% over [^212^Pb]VMT01 in any organ or tissue. This latter finding indicates that free ^208^Tl in the [^212^Pb]VMT01 injectate will not substantially impact estimated tissue absorbed doses in humans.

## 1. Introduction

Melanocortin 1 receptor (MC1R) is a G protein-coupled receptor that is expressed in melanocytes and is implicated in melanogenesis [1]. MC1R is overexpressed on many mouse and human melanoma cells [2,3]. Positron emission tomography (PET) imaging of an MC1R-targeted peptide ^68^Ga-DOTA-GGNle-CycNSH_hex_ in melanoma patients has established clinical proof-of-concept of MC1R as a target for imaging and therapy [4].

Targeted alpha-particle therapy (TAT) is a promising therapeutic strategy that is unique in its ability to deliver cytotoxicity circumventing cellular resistance [5] and has demonstrated significant responses in early clinical trials [6,7,8]. High linear energy transfer (LET) alpha emissions result in clustered DNA double strand breaks [9,10,11,12,13,14,15,16]. In cell culture, alpha emitters have been shown to be more effective in inducing cell death than gamma radiation [17]. Due to short tissue ranges (<100 µm in water, <40 µm in bone), it had previously been believed that TAT may be best suited for the treatment of micrometastases and other disseminated tumors. However, recent TAT studies have demonstrated efficacy in large tumors and there is a growing body of evidence that TAT can activate the immune system and impart both bystander and abscopal effects [18,19]. In the clinical setting, TAT has demonstrated patient benefit even in subjects refractive to beta particle therapy [6].

[^203^Pb]VMT01 is an MC1R-targeted TAT ligand in clinical development (NCT04904120) with elementally matched gamma-emitting [^203^Pb]VMT01 that can be used as an imaging surrogate. [^212^Pb]VMT01 human serum stability and in vivo mouse biodistribution experiments demonstrate robust retention of the ^212^Bi daughter within the chelator following beta emission of parent ^212^Pb and no evidence of in vivo translocation (Li and collaborators, SNMMI-ACNM Mid-Winter Meeting 2022 Abstract) [20]. In addition, the decay physics for ^212^Pb [21,22] (Figure 1) dictates that retention of ^212^Bi within the chelator will subsequently lead to alpha decay via the ^212^Po or ^208^Tl branches at the site of localization due to the short half-lives of ^212^Po (0.3 µs) and ^208^Tl (3.05 m). Due to recoil energy, the 36% alpha emission from ^212^Bi via the ^208^Tl branch will result in the accumulation of free ^208^Tl in the administered injectate. Here, we calculated [^212^Pb]VMT01 human tissue absorbed doses from murine [^203^Pb]VMT01 biodistribution data and the activity and effect of free ^208^Tl in the injectate on tissue absorbed doses.

## 2. Results

### 2.1. Murine [^203^Pb]VMT01 Biodistribution

Murine biodistribution results following intravenous administration of [^203^Pb]VMT01 in female and male CD-1 IGS naïve mice are provided in Appendix A. [^203^Pb]VMT01 cleared rapidly through the kidneys with an accumulation of 6.24 ± 0.35% ID/g and 8.30 ± 1.90% ID/g in females and males, respectively at 0.5 h. Kidney activity decreased to 1.09 ± 0.12% ID/g and 0.55 ± 0.10% ID/g in females and males, respectively at 55 h. Accumulation and retention in other organs were minimal. 

### 2.2. Dosimetry

[^203^Pb]VMT01 and [^212^Pb]VMT01 TIACs (Table 1) and human tissue absorbed doses (Table 2) are provided for a 2 h bladder voiding model. 

For [^203^Pb]VMT01, the tissue with the highest estimated absorbed dose was the urinary bladder wall (0.23 mGy/MBq for females and 0.19 mGy/MBq for males) and the effective dose was 0.028 mSv/MBq and 0.024 mSv/MBq for females and males, respectively. For [^212^Pb]VMT01, the tissue with the highest estimated absorbed dose was the kidneys (8.27 mGy_RBE=5_/MBq for females and 6.83 mGy_RBE=5_/MBq for males). The anticipated dose limiting tissues for [^212^Pb]VMT01 are the red marrow and kidneys, with estimated absorbed doses of 1.06 and 8.27 mGy_RBE=5_/MBq and maximum tolerated activities of approximately 1.9 GBq and 2.2 GBq, respectively, based on published threshold doses from external beam irradiation data [23,24].

Human biodistribution of ^201^Tl chloride published in the literature [25,26] and the calculated activity fraction of free ^208^Tl in the injectate at a shelf-life of 6 h was used to estimate human tissue absorbed doses of administered free ^208^Tl. The activity fraction of free ^208^Tl in the injectate was calculated at a shelf-life of 6 h to be 0.44 MBq ^208^Tl per MBq ^212^Pb (Table 3).

^208^Tl absorbed tissue doses are provided in Table 4. The estimated percent increase in absorbed tissue doses from free ^208^Tl in the injectate was 0.03% and 0.09% in the red marrow and kidneys, respectively. In addition, absorbed doses from free ^208^Tl result in a percent increase of less than 1.2% over [^212^Pb]VMT01 in any organ or tissue, and were within the values that would be expected to be the uncertainty in absorbed dose estimates for [^212^Pb]VMT01 alone.

## 3. Discussion

^212^Pb is a promising alpha-emitting isotope with an elementally matched gamma-emitting isotope ^203^Pb that can be used as an imaging surrogate via single photon emission computed tomography (SPECT). ^212^Pb physical half-life (10.64 h) is attractive from a clinical translation perspective with regard to patient care and waste management. A recently published phase 1 dose escalation trial of targeted alpha therapy with ^212^Pb-DOTAMTATE demonstrated patient safety and promising preliminary efficacy in patients with somatostatin receptor-positive neuroendocrine tumors [27].

From a toxicity standpoint, recoil energy from the emission of an alpha particle decouples the daughter nuclide from any chelator or other chemical bond, and untargeted daughter nuclides are known to accumulate in normal tissues, such as in bone or kidneys [28]. In the work presented here, we calculated estimated human tissue absorbed doses for [^212^Pb]VMT01 from preclinical murine biodistribution data. In addition, we calculated estimated human tissue absorbed doses for free ^208^Tl (that will accumulate in the injectate prior to administration). 

One limitation in the dosimetry of alpha radiotherapeutics is the unknown RBE value. Here, according to the method published by dos Santos and collaborators [21], an RBE value of 5 was used for ^212^Pb alpha emissions and a value of 1 was used for beta and gamma radiation. Recent studies performed in mammary carcinoma NT2.5 cells treated with ^212^Pb-labeled anti-HER2 antibody reported an RBE of 8.3 at 37% survival [29]. Notably, the dose contribution of extracellular unbound ^212^Pb-labeled antibody to the absorbed dose was about 2 orders of magnitude smaller compared to the bound and internalized ^212^Pb, suggesting that extracellular ^212^Pb delivers minimal radiation to cells. The authors conclude that these findings suggest that the actual lesion to dose-limiting tissue absorbed dose could be an order of magnitude greater than that predicted by the calculated absorbed dose. 

The analysis presented here demonstrates that accumulated ^208^Tl in the injectate results in about 1% increase or less in estimated tissue absorbed doses over those projected for [^212^Pb]VMT01. The dosimetry projections for [^212^Pb]VMT01 from [^203^Pb]VMT01 biodistribution data assume that the time-integrated activity coefficient of [^212^Pb]VMT01 applies to all daughter radionuclides. This assumption is valid if there is no in vivo translocation of daughters. Human serum stability and in vivo mouse biodistribution studies demonstrate that ^212^Pb and ^212^Bi remain stably chelated to VMT01 with no evidence of daughter translocation in vivo (Li and collaborators, SNMMI-ACNM Mid-Winter Meeting 2022 Abstract) [20]. Retention of ^212^Pb daughter ^212^Bi within the chelator will result in decay of subsequent daughters ^212^Po and ^208^Tl at the site of localization due to their short half-lives. Prior to administration, accumulation of unchelated ^208^Tl will occur in the formulated product due to the recoil energy of the alpha decay from ^212^Bi. Accumulation of unchelated ^212^Po may also occur prior to administration as a result of beta decay from ^212^Bi; this decay has not yet been characterized. However, due to the extremely short half-life of the ^212^Po daughter (0.3 µs), decay from any free ^212^Po in the intravenously administered product can be assumed to occur in the plasma with negligible radiation to blood cells [29]. 

## 4. Materials and Methods 

### 4.1. Radiolabeling and In Vivo Biodistribution

^203^Pb chloride was obtained from Lantheus Medical Imaging (North Billerica, MA, USA). The structure of VMT01 has been previously published by Li and collaborators [30]. Radiolabeling of VMT01 with ^203^Pb was performed as previously described [30]; radiochemical purity was > 99% as assessed by radio-HPLC. Thirteen-week-old male and female CD-1 IGS mice obtained from Charles River Laboratories (Wilmington, MA, USA) (*n* = 28 per sex, *n* = 56 total) were injected intravenously with [^203^Pb]VMT01 (1.5 ± 0.38 pmol, 74 kBq). Following dosing, animals were sacrificed at 0.5, 1, 2, 4, 6, 24, or 55 h post-injection (*n* = 4 per time point per sex); at each time point whole blood, thymus, thyroid, adrenals, heart, lungs, spleen, bone (femur mid-diaphysis), bone marrow, liver, gallbladder, kidneys (adrenals removed), bladder wall, large intestine (wall and contents), cecum (with contents), small intestines (wall and contents), stomach (wall and contents), pancreas, brain, eyes, skin, muscle (quadriceps), ovaries, testes, uterus, tail, and remaining carcass (at select time points) were resected and assayed for radioactive content by gamma counting. Urine and feces were evaluated for radioactive content using pooled samples from cages. 

### 4.2. Ex Vivo Gamma Counting

The activity of each collected tissue was measured on a Wizard 1480 (Perkin Elmer Life and Analytical Sciences, Bridgeport, CT, USA) or Wizard 2470 (Perkin Elmer Life and Analytical Sciences, Bridgeport, CT, USA) with a 279 keV peak position and 68% window coverage in units of counts per minute (CPM). Triplicate aliquots of the radiotracer, pulled from the dose-calibrated bulk injectate prepared fresh on each day of injections, were weighed, and assayed via gamma counting to convert CPM to units of grams of injected material. The uptake (percent of the injected dose, % ID) and concentration (% ID per gram, % ID/g) were calculated for each sample count using the known injected dose mass, corrected for tail uptake. Concentration estimates used the sample weight of the gamma-counted tissue in grams (g). 

### 4.3. ^203^Pb Dosimetry Analysis 

The radioactivity concentration of [^203^Pb]VMT01 in each organ (fraction of injected activity per gram) over time was used to compute time-integrated activity coefficients (TIAC) [31] for each organ. For all organs except the total body and blood, uptake at time zero was assumed to be 0% ID. Total body and blood were assumed to be 100% ID at time zero. Human TIAC values were defined by multiplying individual mouse concentration values by animal body weight and by the human phantom organ weight to body weight ratio. This method is equivalent to the percent kilogram per gram method [32]. The human phantom organ weight to body weight ratios were determined from the ICRP 89 adult male and adult female phantom organ and total body weights from OLINDA/EXM 2.0 (Hermes Medical Solutions, Stockholm, Sweden). Each time point value was computed from the group average of the data. 

TIAC through the last experimental time point was generated using trapezoidal integration of the seven data points. The contribution to the TIAC following the last experimental time point was estimated by fitting decay-corrected data to a single or a bi-exponential model to estimate biological clearance or assuming physical decay only following the last time point. The combination of physical decay and biological clearance was then analytically integrated. Human TIAC values were then adjusted for radioactivity leaving the body via the renal and gastrointestinal (GI) systems using the dynamic voiding bladder [33] (2 h void) and human alimentary tract model [34]. Excreted urine activity at each time point was defined as 100%-total body % ID-feces % ID. The fraction of excreted urine activity and the voiding half-life were determined by fitting the data to an exponential function. These coefficients were used with a 2 h human voiding time to calculate the urinary bladder TIAC. The ICRP 100 human alimentary tract (HAT) model [34] was utilized with the assumption that radioactivity enters the GI tract via the small intestine. For all animals in each sex group, the radioactivity (% ID, decay corrected) within the small and large intestine, cecum, and all contents were summed at each time point. The peak sum across time for each sex were then determined and used as input into the HAT model in OLINDA/EXM 2.0 to calculate the small intestines, left colon, right colon, and rectum TIACs. Total body radioactivity was calculated as the sum of all measured tissues except for bladder wall, urine, GI, and feces. Total body % ID human was assumed to be equivalent to total body % ID in mouse. The remainder of body TIAC was calculated by subtracting source organ TIACs except for excreta and those derived from the voiding and HAT models. Cortical and trabecular bone TIACs were calculated based on relative surface densities assuming radioactivity distributed to the bone surface. TIAC values were used to compute tissue absorbed dose values for the human adult male and female using OLINDA/EXM 2.0 with ICRP 89 adult male and female phantoms. 

### 4.4. ^212^Pb Dosimetry Analysis 

[^203^Pb]VMT01 data were extrapolated to [^212^Pb]VMT01 by adjusting the radioactive decay half-life. Assuming transient equilibrium between ^212^Pb and its daughters (^212^Bi, ^212^Po, and ^208^Tl), the same residence times as for ^212^Pb were applied to the daughter nuclides as described by dos Santos and collaborators [21] OLINDA/EXM 2.0 calculations were performed for all nuclides manually. For ^208^Tl and ^212^Po, the relevant branching fraction was applied. A relative biological effectiveness (RBE) value of 5 was used for the alpha emissions from ^212^Bi and ^212^Po (while an RBE of 1 was used for beta and gamma emissions); absorbed doses are presented in units of Gray (Gy_RBE=5_).

### 4.5. ^208^Tl Dosimetry Analysis

Human biodistribution of ^201^Tl chloride (a cardiac imaging agent) via scintigraphy imaging published in the literature [25,26] was used to estimate the dosimetry of free ^208^Tl. Thallous ion behaves as a potassium analog and tissue uptake is essentially intracellular. Biodistribution of thallium at early times in organs is thus related to regional blood flow. Human % ID values for heart, brain, kidney, liver, intestine, spleen, testes, and the remainder of body were determined from scintigraphy imaging as reported by Svensson and collaborators [26] and Krahwinkel and collaborators [25] for ^201^Tl chloride (using the earliest imaging time point from a combination of at rest and after exercise) and conservatively assuming no biological clearance and 100% ID in the total body (Table 5). Radioactive decay of ^208^Tl (3.05 m half-life) and resulting TIAC values were used to determine tissue-absorbed doses in the ICRP 89 human adult male using OLINDA/EXM 2.2. The activity fraction of free ^208^Tl in the injectate at a shelf-life of 6 h was calculated using the ^212^Pb decay scheme and branching fraction of 35.94% for ^208^Tl. The ^208^Tl activity fraction was used to calculate the ^208^Tl mGy_RBE=5_/MBq administered ^212^Pb activity.

## 5. Conclusions

The critical tissues for [^212^Pb]VMT01 based on human dosimetry estimates from murine [^203^Pb]VMT01 biodistribution data and tissue threshold doses from external beam irradiation data are anticipated to be red marrow and kidneys. Dosimetry analysis indicates that free ^208^Tl that will accumulate in the [^212^Pb]VMT01 injectate prior to administration will not substantially impact estimated tissue absorbed doses in humans. The dosimetry estimations support the clinical evaluation of [^212^Pb]VMT01. 

## Figures and Tables

**Figure 1 molecules-27-05831-f001:**
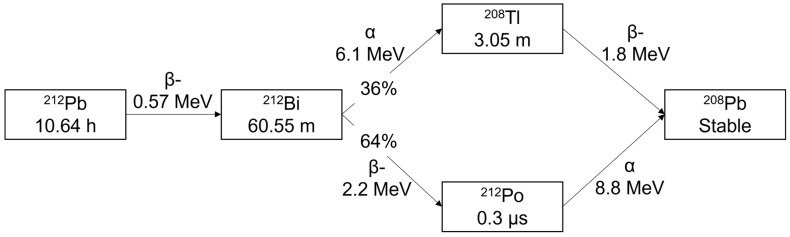
^212^Pb decay scheme [21,22].

**Table 1 molecules-27-05831-t001:** [^203^Pb]VMT01 and [^212^Pb]VMT01 time-integrated activity coefficients.

Organ	[^203^Pb]VMT01 TIAC (MBq h/MBq)		[^212^Pb]VMT01 TIAC (MBq h/MBq)	
	Female	Male	Female	Male
Adrenal glands	5.08 × 10^−5^	3.72× 10^−4^	4.75 × 10^−5^	3.33 × 10^−4^
Brain	3.53 × 10^−4^	1.55× 10^−3^	3.06 × 10^−4^	1.24 × 10^−3^
Cortical bone	1.50 × 10^−2^	8.89 × 10^−2^	1.20 × 10^−2^	6.34 × 10^−2^
Eyes	4.24 × 10^−5^	8.03 × 10^−5^	3.71 × 10^−5^	6.73 × 10^−5^
Gallbladder	2.07 × 10^−4^	2.22 × 10^−4^	1.45 × 10^−4^	1.84 × 10^−4^
Heart contents	2.28 × 10^−2^	2.62 × 10^−2^	2.22 × 10^−2^	2.59 × 10^−2^
Heart wall	9.93 × 10^−4^	1.91 × 10^−3^	7.74 × 10^−4^	1.43 × 10^−3^
Kidneys	2.00 × 10^−1^	1.48 × 10^−1^	9.70 × 10^−2^	9.01 × 10^−2^
Left colon	3.49 × 10^−1^	4.26 × 10^−1^	1.03 × 10^−1^	1.51 × 10^−1^
Liver	1.09 × 10^−1^	8.85 × 10^−2^	4.29 × 10^−2^	3.68 × 10^−2^
Lungs	1.79 × 10^−2^	2.18 × 10^−2^	1.11 × 10^−2^	1.66 × 10^−2^
Ovaries	1.46 × 10^−4^	-	4.99 × 10^−5^	-
Pancreas	3.68 × 10^-4^	1.19 × 10^−3^	2.95 × 10^−4^	9.27 × 10^−4^
Rectum	2.87 × 10^−1^	3.67 × 10^−1^	5.04 × 10^−2^	8.47 × 10^−2^
Red marrow	1.58 × 10^−1^	2.24 × 10^−4^	1.39 × 10^−3^	1.79 × 10^−4^
Right colon	4.23 × 10^−1^	4.94 × 10^−1^	2.10 × 10^−1^	2.69 × 10^-1^
Small intestines	1.28 × 10^−1^	1.91 × 10^−1^	1.07 × 10^−1^	1.60 × 10^−1^
Spleen	3.00 × 10^−3^	3.51 × 10^−3^	1.52 × 10^−3^	1.86 × 10^−3^
Stomach contents	1.09 × 10^−2^	1.84 × 10^−2^	7.28 × 10^−3^	7.34 × 10^−3^
Testes	-	3.85 × 10^−4^	-	2.85 × 10^−4^
Thymus	6.52 × 10^−5^	1.41 × 10^−4^	5.29 × 10^−5^	1.25 × 10^−4^
Thyroid	1.38 × 10^−4^	4.69 × 10^−4^	1.05 × 10^−4^	3.41 × 10^−4^
Total body/remainder	4.55 × 10^0^	9.20 × 10^−1^	1.18 × 10^0^	8.83 × 10^−1^
Trabecular bone	1.50 × 10^−2^	8.89 × 10^−2^	1.20 × 10^−2^	6.34 × 10^−2^
Urinary bladder	1.49 × 10^0^	1.39 × 10^0^	1.40 × 10^0^	1.31 × 10^0^
Uterus	1.05 × 10^−3^	-	7.46 × 10^−4^	-

**Table 2 molecules-27-05831-t002:** Human tissue absorbed doses.

Organ/tissue	^203^Pb Absorbed Dose (mGy/MBq)		^212^Pb Absorbed Dose (mGy_RBE=5_/MBq)	
	Female	Male	Female	Male
Adrenals	1.14 × 10^−2^	1.11 × 10^−2^	1.06 × 10^−1^	5.83 × 10^−1^
Brain	2.17 × 10^−3^	5.86 × 10^−4^	7.88 × 10^−3^	2.20 × 10^−2^
Breasts	6.71 × 10^−3^	-	4.67 × 10^−1^	-
Oesophagus	7.55 × 10^−3^	3.12 × 10^−3^	4.69 × 10^−1^	2.90 × 10^−1^
Eyes	3.78 × 10^−3^	1.02 × 10^−3^	6.16 × 10^−2^	1.08 × 10^−1^
Gallbladder wall	1.84 × 10^−2^	1.14 × 10^−2^	4.86 × 10^−1^	3.07 × 10^−1^
Left colon	9.83 × 10^−2^	1.16 × 10^−1^	8.31 × 10^−1^	8.46 × 10^−1^
Small intestine	2.87 × 10^−2^	2.58 × 10^−2^	5.96 × 10^−1^	4.40 × 10^−1^
Stomach wall	1.27 × 10^−2^	6.97 × 10^−3^	4.83 × 10^−1^	3.02 × 10^−1^
Right colon	7.40 × 10^−2^	8.24 × 10^−2^	8.50 × 10^−1^	8.07 × 10^−1^
Rectum	1.18 × 10^−1^	1.17 × 10^−1^	7.35 × 10^−1^	6.39 × 10^−1^
Heart wall	7.38 × 10^−3^	4.45 × 10^−3^	7.86 × 10^−1^	7.06 × 10^−1^
Kidneys	4.23 × 10^−2^	2.80 × 10^−2^	8.27 × 10^0^	6.83 × 10^0^
Liver	1.15 × 10^−2^	7.05 × 10^−3^	7.32 × 10^−1^	4.91 × 10^−1^
Lungs	6.59 × 10^−3^	2.82 × 10^−3^	2.81 × 10^−1^	3.30 × 10^−1^
Ovaries	2.60 × 10^−2^	-	1.59 × 10^−1^	-
Pancreas	1.12 × 10^−2^	1.15 × 10^−2^	7.45 × 10^−2^	1.76 × 10^−1^
Prostate	-	2.53 × 10^−2^	-	3.46 × 10^−1^
Salivary glands	7.27 × 10^−3^	1.54 × 10^−3^	4.66 × 10^−1^	2.87 × 10^−1^
Red Marrow	1.06 × 10^−3^	5.46 × 10^−3^	8.64 × 10^−1^	1.06 × 10^0^
Osteogenic Cells	1.46 × 10^−2^	1.23 × 10^−2^	3.88 × 10^0^	6.95 × 10^0^
Spleen	1.16 × 10^−2^	6.11 × 10^−3^	2.90 × 10^−1^	3.01 × 10^−1^
Testes	-	5.26 × 10^−3^	-	2.07 × 10^−1^
Thymus	5.47 × 10^−3^	1.94 × 10^−3^	6.90 × 10^−2^	1.23 × 10^−1^
Thyroid	4.88 × 10^−3^	2.16 × 10^−3^	1.50 × 10^−1^	4.04 × 10^−1^
Urinary bladder wall	2.29 × 10^−1^	1.89 × 10^−1^	2.95 × 10^0^	2.14 × 10^0^
Uterus	4.83 × 10^−2^	-	3.27 × 10^−1^	-

**Table 3 molecules-27-05831-t003:** ^212^Pb, ^212^Bi, and ^208^Tl activity and activity fraction in injectate preparation at 0 h and 6 h for nominal 1 MBq ^212^Pb-VMT01.

	0 h		6 h	
	Activity (MBq)	Activity Fraction	Activity (MBq)	Activity Fraction
^212^Pb	1.00	1.00	0.68	1.00
^212^Bi	0.00	0.00	0.73	1.08
^208^Tl	0.00	0.00	0.29	0.44

**Table 4 molecules-27-05831-t004:** Tissue absorbed dose estimates for human adult male for free ^208^Tl in the injectate at a shelf-life of 6 h, [^212^Pb]VMT01 human adult male, total absorbed dose, and % increase in absorbed dose from free ^208^Tl contribution.

Organ/Tissue	^208^Tl Absorbed Dose (mGy/MBq)	[^212^Pb]VMT01 Absorbed Dose (mGy_RBE=5_/MBq)	Total Absorbed Dose (mGy_RBE=5_/MBq)	^208^Tl % Increase
Adrenals	3.09 × 10^−3^	5.83 × 10^−1^	5.84 × 10^−1^	0.23
Brain	5.98 × 10^−4^	2.20 × 10^−2^	2.23 × 10^−2^	1.18
Oesophagus	9.40 × 10^−4^	2.90 × 10^−1^	2.90 × 10^−1^	0.14
Eyes	5.09 × 10^-4^	1.08 × 10^−1^	1.08 × 10^−1^	0.21
Gallbladder wall	1.57 × 10^−3^	3.07 × 10^−1^	3.08 × 10^−1^	0.22
Left colon	7.38 × 10^−3^	8.46 × 10^−1^	8.49 × 10^−1^	0.38
Small intestine	7.60 × 10^−3^	4.40 × 10^−1^	4.43 × 10^−1^	0.75
Stomach wall	1.14 × 10^−3^	3.02 × 10^−1^	3.02 × 10^−1^	0.16
Right colon	7.06 × 10^−3^	8.07 × 10^−1^	8.10 × 10^−1^	0.38
Rectum	6.58 × 10^−3^	6.39 × 10^−1^	6.42 × 10^−1^	0.45
Heart wall	3.62 × 10^−3^	7.06 × 10^−1^	7.08 × 10^−1^	0.22
Kidneys	1.41 × 10^−2^	6.83 × 10^0^	6.84 × 10^0^	0.09
Liver	1.97 × 10^−3^	4.91 × 10^−1^	4.92 × 10^−1^	0.17
Lungs	8.02 × 10^−4^	3.30 × 10^−1^	3.30 × 10^−1^	0.11
Pancreas	1.73 × 10^−3^	1.76 × 10^−1^	1.77 × 10^−1^	0.43
Prostate	1.02 × 10^−3^	3.46 × 10^−1^	3.46 × 10^−1^	0.13
Salivary glands	5.98 × 10^−4^	2.87 × 10^−1^	2.87 × 10^−1^	0.09
Red marrow	7.70 × 10^−4^	1.06 × 10^0^	1.06 × 10^0^	0.03
Osteogenic cells	6.88 × 10^−4^	6.95 × 10^0^	6.95 × 10^0^	0.00
Spleen	3.24 × 10^−3^	3.01 × 10^−1^	3.02 × 10^−1^	0.47
Testes	3.59 × 10^−3^	2.07 × 10^−1^	2.09 × 10^−1^	0.75
Thymus	8.71 × 10^−4^	1.23 × 10^−1^	1.23 × 10^−1^	0.31
Thyroid	6.39 × 10^−4^	4.04 × 10^−1^	4.04 × 10^−1^	0.07
Urinary bladder wall	8.85 × 10^−4^	2.14 × 10^0^	2.14 × 10^0^	0.02

**Table 5 molecules-27-05831-t005:** ^208^Tl human tissue % ID.

Organ/Tissue	Human % ID
Heart	3.2 [26]
Brain	1.5 [25]
Kidneys	12.5 [26]
Liver	5.1 [25]
Intestine ^a^	20.1 [25]
Spleen	1.0 [25]
Testes	0.4 [25]
Remainder of body	56.2 [25]

^a^ Activity was split equally between the small intestine, upper large intestine wall, lower large intestine wall, and rectum wall based on ICRP 89 target wall organ masses.

## Data Availability

The data presented in this study may be available on request from the corresponding author.

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
