# Peer review of "Dosimetry of [212Pb]VMT01, a MC1R-Targeted Alpha Therapeutic Compound, and Effect of Free 208Tl on Tissue Absorbed Doses"

_molecules, 2022, doi:10.3390/molecules27185831_

Round 1
Reviewer 1 Report
Nice and well written paper with interesting and useful results and conclusions in the growing and important field of therapeutic use of radiopharmaceuticals.
The paper can be improved on the basis of following comments:
- Many readers are not familiar with the radiation characteristics/physics of lead-212. A decay scheme of lead-212 with the different daughter (radio) nuclides, their half-life and the probality of the decay branches will be helpful in the introduction of the paper.
- In the Materials and Methods section, the source/origin of lead-212 and of Pb-212 VMT01 and of Pb-203 VMT01 is missing. Were they synthesized in-house? If so, include a reference to a paper in which the radiosynthesis is described. Add also the radiochemical purity, as this is important for the radiation dosimetry. If they were provided by an external lab/supplier, provide this information.
- Reference must be made to a paper in which the structure of Pb-VMT01 is shown (probably Cancers 2021, 13, 3676 ) in order to show to the reader that the chelator is DOTA, which forms stable complexes with lead, bismuth and polonium.
- In the Materials and Methods section, the city and state or country of the companies mentioned must be added.
- In a scientific paper IS units must be used. The SI unit of radioactivity is the becquerel and no longer curie. So, change all millicurie or microcuries mentions to the corresponding Bq, kBq or Mbq values and units (unless the mCi value is given within brackets after the becquerel value)
- Animals can be sacrificed only once, so the 3rd line of the Materials and Methods chapter should read: animals were sacrificed at 0.5, 1, 2, 4, 6, 24, and or 55 h post-injection.
- In Table 1 Kidney should read Kidneys
- The title of Molecular Pharmaceutics (in reference of submitted paper) must be abbreviated to Mol Pharm.
- On page 2, the sentence ‘the decay physics for 212Pb dictates that stability of 212Bi will subsequently lead to…’ is incorrect. 212Bi is not stable (as it decays) but the 212Bi-chelate is (probably) stable.
- In the introduction chapter, a sentence should be added to clarify to the reader that VMT01 labeled with lead-203, decaying by electron capture and gamma radiation, can be used instead of the lead-212 labeled tracer agent for biodistribution and radiation dosimetry studies. It is now mentioned in the Discussion chapter, but the reader should be informed earlier, before he/she reads the Materials and the Results chapters.
- On page 2: quadricep = quadriceps
- Spelling should be checked. Isn’t it body weight instead of bodyweight and time point instead of timepoint?
- There should not be a dot after the title of a Table, unless the title is a sentence with a verb (Table 1 and Table 3).
- Two different tables have the title Table 1. Adjust.
- Percent and percentage are not correctly used. The rule for using percent and percentage is straightforward. The word percent (or the symbol %) accompanies a specific number, whereas the more general word percentage is used without a number.
Author Response
Reviewer 1:
- Many readers are not familiar with the radiation characteristics/physics of lead-212. A decay scheme of lead-212 with the different daughter (radio) nuclides, their half-life and the probability of the decay branches will be helpful in the introduction of the paper.
We have added the 212Pb decay scheme. Please see addition of Figure 1 in the revised manuscript.
- In the Materials and Methods section, the source/origin of lead-212 and Pb-212 VMT01 and of Pb-203 VMT01 is missing. Were they synthesized in-house? If so, include a reference to a paper in which the radiosynthesis is described. Add also the radiochemical purity, as this is important for the radiation dosimetry. If they were provided by an external lab/supplier, provide this information.
Yes, 203Pb-VMT01 was synthesized in house. A reference describing the radiosynthesis has been added to the revised manuscript and the source of 203Pb and radiochemical purity details have also been added.
- Reference must be made to a paper in which the structure of Pb-VMT01 is shown (probably Cancers 2021, 13, 3676) in order to show to the reader that the chelator is DOTA, which forms stable complexes with lead, bismuth and polonium.
We have added the reference to the Cancers 2021 publication which does indeed show the structure.
- In the Materials and Methods section, the city and state or country of the companies mentioned must be added.
City and state or country of companies has been added to the Materials and Methods.
- In a scientific paper IS units must be used. The IS unit of radioactivity is the becquerel and no longer curie. So, change all millicurie or microcuries mentions to the corresponding Bq, kBq or MBq values and units (unless the mCi value is given within brackets after the becquerel value)
All radioactivity units have been changed to MBq.
- Animals can be sacrificed only once, so the 3rd line of the Materials and Methods chapter should read: animals were sacrificed at 0.5, 1, 2, 4, 6, 24 and or 55 h post-injection.
Corrected
- In Table 1 Kidney should read Kidneys
Corrected
- The title of Molecular Pharmaceutics (in reference of submitted paper) must be abbreviated to Mol Pharm.
This reference has been changed to an accepted conference abstract.
- On page 2, the sentence “the decay physics for 212Pb dictates that stability of 212Bi will subsequently lead to…” is incorrect. 212Bi is not stable (as it decays) but the 212Bi-chelate is (probably) stable.
Sentence has been edited to change the phrasing to “retention of 212Bi within the chelate”. Several other edits have also been made to the introduction to change the word stability to retention with respect to chelated 212Bi.
- In the introduction chapter, a sentence should be added to clarify to the reader that VMT01 labeled with lead-203, decaying by electron capture and gamma radiation, can be used instead of the lead-212 labeled tracer agent for biodistribution and radiation dosimetry studies. It is now mentioned in the Discussion chapter, but the reader should be informed earlier, before he/she reads the Materials and the Results chapters.
We have edited the first sentence of the third paragraph of the introduction to specify “…elementally matched gamma-emitting [203Pb]VMT01 that can be used as an imaging surrogate.”
- On page 1: quadricep = quadriceps
Corrected
- Spelling should be checked. Isn’t it body weight instead of bodyweight and time point instead of timepoint.
Corrected
- There should not be a dot after the title of a Table, unless the title is a sentence with a verb (Table 1 and Table 3).
Corrected
- Percent and percentage are not correctly used. The rule for using percent and percentage is straightforward. The word percent (or the symbol %) accompanies a specific number, whereas the more general word percentage is used without a number.
Corrected
Reviewer 2 Report
Authors reported the dosimetry evaluation of a MC1R-targeted alpha-radiotracer 212Pb-VMT01 and the effect of Tl-208 on the absorbed radiation dose. The manuscript was well-written, and experiments were well-performed. The results from this study is critically important for those who are interested in the development of Pb-212-based radiotracers and have concerns on the presence of Tl-208. Thus, it is recommended for publication for this journal.
Comments:
1. It would be better to combine this with reference #20 for a much "complete study". However, this manuscript in its current form is excellent already.
2. It is highly recommended to give more schematic description on Pb-212 decay in the introduction section or early discussion section.
Author Response
Reviewer 2:
- It would be better to combine this with reference #20 for a much “complete study.” However, this manuscript in its current form is excellent already.
Thank you very much for the comment. We believe the combined work with sufficient detail may be too long for a single manuscript.
- It is highly recommended to give more schematic description on Pb-212 decay in the introduction section or early discussion section.
We have added the 212Pb decay scheme. Please see addition of Figure 1 in the revised manuscript.